# The Effect of Wheatgrass Lyophilizate on Blood Clotting Time in Rats

**István Takács** [1], **Melinda Urkon** [1,2], **Tivadar Kiss** [1,3], **Edina Horváth** [4], **Eszter Laczkó-Zöld** [2], **Zoltán Péter Zomborszki** [1,3], **Anita Lukács** [3,4,5], **Gábor Oszlánczi** [3,4], **Dezső Csupor** [1,3,6,*] **and Andrea Szabó** [3,4]

1   Department of Pharmacognosy, Faculty of Pharmacy, University of Szeged, 6 Eötvös Street,
    H-6720 Szeged, Hungary; taki.biotech@gmail.com (I.T.); urkonmelinda1@gmail.com (M.U.);
    kiss.tivadar@pharmacognosy.hu (T.K.); zombozope@pharmacognosy.hu (Z.P.Z.)
2   Department of Pharmacognosy and Phytotherapy, George Emil Palade University of Medicine, Pharmacy,
    Science, and Technology of Târgu Mureş, Gheorghe Marinescu Street 38, 540139 Târgu Mureş, Romania;
    zoldeszter@yahoo.com
3   Interdisciplinary Centre for Natural Products, University of Szeged, 6 Eötvös Street, H-6720 Szeged, Hungary;
    lukacs.anita@med.u-szeged.hu (A.L.); oszlanczi.gabor@gmail.com (G.O.);
    szabo.andrea@med.u-szeged.hu (A.S.)
4   Department of Public Health, Faculty of Medicine, University of Szeged, H-6720 Szeged, Hungary;
    korosine.edina@med.u-szeged.hu
5   Department of Physiology, Anatomy and Neuroscience, Faculty of Science and Informatics,
    University of Szeged, Közép Fasor 52, H-6726 Szeged, Hungary
6   Department of Clinical Pharmacy, Faculty of Pharmacy, University of Szeged, Szikra Street 8,
    H-6725 Szeged, Hungary
*   Correspondence: csupor.dezso@pharmacognosy.hu; Tel.: +36-62-54-5559

**Abstract:** Wheatgrass is widely used in the alternative medicine, however, there is a lack of clinical evidence to support its efficacy. Although based on its chemical composition, data from animal experiments and clinical trials, the use of juice and extracts of *Triticum* shoots seems to be safe, clinical reports point out its potential interaction with oral anticoagulants. The aim of our study was to assess the interaction of wheatgrass with warfarin in rats and to assess its flavonoid content. Three groups of animals were treated orally with wheatgrass, warfarin, or the combination of wheatgrass and warfarin for five days. Clotting assays were performed using platelet-poor plasma. Prothrombin time was determined by optical and mechanical coagulometers. Flavonoid content of wheatgrass was measured by HPLC. The effect of wheatgrass on prothrombin time was not confirmed. Co-administration of wheatgrass and warfarin did not result in diminished anticoagulant activity. Low amount of flavonoids was detected in wheatgrass juice, the total flavonoid content was 0.467 mg/100 g lyophilized juice powder. The previously reported rutin, quercetin and apigenin was not detected by us. Our results do not confirm the probability of interaction of wheatgrass with oral anticoagulants. However, the low flavonoid content of wheatgrass does not support its use as an antioxidant.

**Keywords:** blood clotting; flavonoid; *Triticum*; warfarin; wheatgrass

## 1. Introduction

Wheatgrass is one of the most versatilely used plants in contemporary folk medicine, with indications ranging from diabetes to cancer. The medicinal application of young cereal grass leaves—harvested just before the jointing stage—started at the beginning of the 20th century, after the discovery of structural similarities of hemoglobin and chlorophyll. In 1927, Charles Schnabel, an agricultural chemist reasoned that green vegetables would build blood and therefore may be used as a growth promoter [1]. Interestingly, the structural similarities of chlorophyll and hemoglobin and the potential consequences in bioactivity, even the "limited use of chlorophyll as a blood substitute" have been discussed even in

recent papers [2,3]. Some articles claim that the wheatgrass was used as a health tonic in folk and Ayurvedic medicine to increase the hemoglobin level in different types of anemic patients [4], however, these seem to be dubious, since increasing hemoglobin level was obviously out of the scope in Ayurvedic medicine. Nevertheless, some clinical studies aimed at investigating this supposed effect. In one study, wheatgrass treatment increased the quality of life of children with thalassemia, however there was no reduction in the frequency of blood transfusion requirement [5].

The first commercial product made from the young grass shoots from wheat, barley, and rye (Cerophyl) achieved great financial success, and beside agricultural use, it was applied in human therapy as well. First, it was used as multivitamin, later for more specific therapeutic indications. According to case reports, it has increased the prothrombin index of patients suffering from hemorrhage with jaundice, and this effect was attributed to the supposed vitamin K content of the product (or to the presence of other unknown factors) [6]. In a randomized double-blind placebo-controlled trial, daily consumption of 100 mL wheatgrass juice significantly reduced the overall disease activity index and the severity of rectal bleeding in patients with ulcerative colitis after one month of treatment, however, the latter observation was not associated with the supposed blood clotting effect [7]. The effect of wheatgrass on blood coagulation has not been assessed in modern studies, especially not in humans. However, Tirgar et al. found that wheatgrass significantly increased platelet counts and reduced bleeding and clotting time in busulfan-induced thrombocytopenic rats [8].

Today, wheatgrass (7–8 days germinated young grass of *Triticum* sp.) is marketed as antioxidant, general detoxifier, and for the treatment of several diseases, including cancer [9]. The antioxidant activity might partly be related to the phenolic acid content of wheat [10]. Apart from several in vitro studies, antioxidant effect was confirmed in animal experiments [11] and in a clinical trial as well, where supplementation with wheatgrass provided protection against lipid peroxidation and thereby it decreased oxidative stress [12].

Although many papers refer to the flavonoid content of wheatgrass, the majority of these used non-specific tests to support this hypothesis [13–17]. The presence of rutin [18] and quercetin [19] in wheatgrass were reported based on HPLC experiments. The antimutagenic activity of wheat sprout extract was demonstrated on rats. The authors claim that "inhibitory compounds of wheat sprout extract could be the hydrophilic derivatives of flavonoids, especially apigenin", however, no experimental confirmation was presented to support this hypothesis [20]. An early paper reported the presence of apigenin glycosides primarily based on co-chromatography with test substances and IR-spectroscopic comparison [21]. In a recent review, apigenin was highlighted as the major flavonoid of wheatgrass, with anticarcinogenic, antiproliferative, and proapoptotic effects [22].

Concerning the supposed anticancer effect, the majority of the studies were carried out in vitro. Wheatgrass extract had cytotoxic effect on HL60 [23], HepG2 [24], MCF7, and HeLa cells in vitro, moreover, exerted synergistic effect with cisplatin [25]. The extract decreased the metastatic protein expressions in Hep-2 cells [26]. The in vivo (human) studies focused on the complementary effect of wheatgrass rather than on its antitumor effect. In a study with 60 participants—where daily 60 mL wheatgrass juice was taken during chemotherapy—reduced myelotoxicity, dose reductions, and need for granulocyte colony-stimulating factor (GCSF) support were observed compared to the regular supportive therapy, without affecting the efficacy of chemotherapy [27]. One paper reported that the administration of wheatgrass with chemotherapy reduced the severity of side-effects in patients with leukemia [28].

Early clinical reports refer to the prothrombin index-increasing effect of wheatgrass [6], later studies describe the decrease of rectal bleeding during the treatment [7]. Reduction of clotting time was reported in animal experiments as well [8]. However, the influence on coagulation is not always a desired outcome, for patients taking anticoagulants the potential procoagulant activity of wheatgrass may lead to interaction and serious conse-

quences. Those patients who are on oral anticoagulation therapy are usually informed about interactions with prescription drugs and some foods, but rarely about interactions with medicinal herbs.

Today, warfarin is the most extensively used oral anticoagulant for the prevention and treatment of thromboembolic complications in cardiovascular diseases such as atrial fibrillation, venous thrombosis, and pulmonary embolism [29]. The potential interactions between warfarin and herbal drugs have been studied extensively, however wheatgrass has not been studied from this aspect [30–32].

Considering the widespread use of wheatgrass and the potential wheatgrass-warfarin interactions, our goal was dual: to assess its effect on blood clotting after oral administration in rats, and to measure the flavonoid content of wheatgrass, with special focus on apigenin.

## 2. Materials and Methods

### 2.1. Experimental Substances

Frozen fresh pressed wheatgrass juice (Gyorsfagyasztott Bio Tönköly Búzafűlé) was obtained from commercial sources and lyophilized using Christ Alpha1-4 freeze-drier for 25.5 h (dry-freezing method is shown in Supplementary Table S1). About 1000 g frozen juice yielded 42 g green, light powder. The powder was stored in hermetic bag at room temperature till use. According to the manufacturer the juice was obtained from *Triticum spelta* L. (Poaceae) by mechanical pressing, without the addition of solvents and any other additives.

Warfarin was administered in the form of powdered tablets (Warfarin Orion 3 mg warfarin, Orion Pharma Hungary, LOT: 1689957). Reagent Neoplastine CI 5 was purchased from commercial source (Diagnostica Stago, France). Flavonoids (apigenin, apigenin-7-*O*-glycoside, eupatorin, hesperidin, hyperoside, kaempferol, luteolin, rutin, and quercetin) were previously isolated from different plants at the Department of Pharmacognosy, University of Szeged. The identification and purity (>97%) were determined by means of NMR and HPLC.

### 2.2. Sample Preparation Method for HPLC Analysis

During the preliminary HPLC experiments lyophilized wheatgrass juice was dissolved in methanol. For detailed HPLC analysis, 10 g of lyophilized wheatgrass juice powder (equivalent to 238 g of fresh wheatgrass juice) was fractioned on a 70 g polyamide column (MP Biomedicals Germany GmbH, Hessen, Germany) with mixtures of methanol and water (8:2, 6:4, 4:6, 2:8, 0:1). The volume of each fraction was 1.4 l, respectively. The fractions were concentrated in vacuo and the dry residues were redissolved in methanol for further HPLC analysis.

### 2.3. Flavonoid Content Determination

The flavonoid content of the fractions was analyzed using a HPLC system comprising a Shimadzu LC-20AD pump, DGU-20A5R degasser, SIL20ACH autosampler (tempered to 25 °C), CTO-20AC column oven, and SPD-M20A photodiode array detector modules, connected with CBM-20A control module. Column temperature was set to 35 °C. The solvent system consisted of 0.1% trifluoro-acetic acid in water (A) and methanol (B). Gradient elution was applied (0 min: 20% B, 20 min: 40% B, 24 min: 55% B, 27 min: 70% B, 28 min: 70% B, 28.5 min: 20% B, 34 min: 20% B). For separation, a Kinetex C-18 (100 Å, 250 × 4.6 mm, 5 μm) column was used (Phenomenex, Torrance, CA, USA). The injected volume was 50 μL, flow rate was 1.5 mL/min. Detection was performed in the whole UV wavelength range, and flavonoids were detected at 350 nm. For screening for flavonoid content, the previously reported apigenin, rutin, and quercetin, and some other common flavonoids (apigenin-7-*O*-glycoside, eupatorin, hesperidin, hyperoside, kaempferol, and luteolin) were used. We established calibration curve for apigenin (the flavonoid claimed to be the major representative of this chemical group [22]) based on 6 points. The total flavonoid content was determined by integrating chromatogram peaks in the retention

time range 1–34 min with characteristic flavonoid absorbance bands (band A: 300–385, band B: 250–295) at 350 nm [33].

### 2.4. Animals and Husbandry

About 24 male SPF Wistar male rats were included in this experiment (purchased from Toxi-Coop Zrt., Budapest, Hungary), weighed 230–250 g at the start of the study. After six days of acclimatization the rats were randomized according to body weight into three groups (wheatgrass [WG], warfarin [WF], warfarin and wheatgrass [WW], 8 animals/group). As environmental enrichment we used unbleached, clean paper tubes. Dust-free wood shavings were applied as bedding material. Rats were kept under standard climatic conditions (22–24 °C, 12 h light/dark cycle with light starting at 6:00 a.m., 2–3 rats in one cage) with free access to drinking water. Rats were fed with standard, certified rodent chow.

### 2.5. Warfarin and Wheatgrass Administration

Rats were treated with wheatgrass (WG), warfarin (WF), and the combination of wheatgrass and warfarin (WW). The dose of warfarin was 0.4 mg/kg b.w./day [34]. The 127.5 mg/kg b.w. dosage for rats of lyophilized wheatgrass powder was considered as six-fold of the average recommended amount of frozen wheatgrass juice for human (30.35 g/day frozen juice is equivalent to 21.25 mg/kg b.w. of lyophilized powder in case of 60 kg b.w. adult person) [35]. Self-made sugar cookie dough was applied as vehicle (pelletized to balls) in all groups [36] instead of traditional gavage technique in order to cause the least stress and harm to the animals and to model human exposure the most objectively. All animals received 4 g cookie dough/kg b.w. orally.

During a five-day long acclimatization period all rats were habituated to the vehicle cookie balls. The acclimatization was followed by five days of treatment. The treatments were performed after overnight fasting. After the treatment, free access to standard rodent chow was provided for 3 h. During the 5-day study, the body weight of animals were measured every morning. According to the measured body weight, individual special feed portions were prepared (dough balls incorporating the necessary warfarin and/or wheatgrass powder, e.g., if the body weight of a rat in WW group was 252 g, it received 0.1008 mg warfarin and 32.13 mg wheatgrass in 1.008 g dough ball). In order to minimize stress and avoid refraining from eating, the balls were given to each animal individually in the morning hours between 10 and 11 a.m., while they were left in their home cage. The feeding performance was 100% during the treatment period; no uneaten cookie dough was noticed.

### 2.6. Blood Samples

Blood samples were collected before the beginning of the 5-day long treatment from the tail veins of awaken but restricted rats, and after the treatment period—to be sure to get the maximum amount of blood—from abdominal vein. For this latter, all animals were over-anaesthetized (euthanized by a widely accepted way) with isoflurane inhalation by using a precision vaporizer and administering the isoflurane gas gently up to >5% in 100% oxygen that continued until lack of respiration for at least 1 min. The citrated blood samples (9 parts blood 1 part 3.8% trisodium citrate) were centrifuged at 2700 G and the platelet-poor plasma was used for further investigations.

### 2.7. Clotting Assays

Clotting assays were performed using platelet-poor plasma. Prothrombin time was determined using two parallel measurements, by optical and mechanical coagulometers. For the experiment, 50 μL of plasma was incubated for 2 min at 37 °C, and then 100 μL of Neoplastine CI 5 (Diagnostica Stago, Asnières-sur-Seine, France) was added.

TEChrom IV plus manual 4-channel coagulation analyzer equipped with Hemostasis Analyzer (Teco GmbH, Neufahrn, Germany) was used as the optical coagulometer. The clot

formation was determined by measuring the absorbance 1 min after adding Neoplastine CI 5 on 405 nm wavelength. A mechanical coagulometer Merlin MC 1 (Merlin Medical, ABW Medizin und Technik, Lemgo, Germany) equipped with tempered incubation block on constant temperature at 37 °C was also used to determine the prothrombin time.

*2.8. Statistical Analysis*

The distribution of data was checked for normality by Shapiro–Wilk test, and Kruskal–Wallis test was used for evaluation. In case of significance ($p < 0.05$) the data were tested using the Mann–Whitney test, to show which groups are significantly different from each other. Statistical analyses were carried out using R (version 3.5.1, The R Foundation for Statistical Computing, Vienna, Austria, https://www.r-project.org/ accessed on 10 August 2021).

**3. Results**

One of the main aims of this study was to quantitatively analyze the flavonoid content of a pressed wheatgrass juice. For preliminary analysis, the juice was lyophilized and redissolved in methanol. The whole chromatogram was scanned for peaks with UV spectra characteristic to flavonoids. No flavonoids could be detected in this methanolic solution by HPLC-DAD based on their retention times and UV spectra. Therefore, the lyophilized juice was fractioned by column chromatography on polyamide stationary phase in order to enrich flavonoids in certain fractions (eluted with more apolar solvents) and to separate from chlorophyll. The presence of flavonoids was screened in all the five fractions (see Section 2.2). The presence of screened flavonoids (i.e., apigenin, apigenin-7-*O*-glycoside, eupatorin, hesperidin, hyperoside, kaempferol, luteolin, rutin, and quercetin; chromatogram of the reference compounds see in Figure S1) could not be confirmed in any of the fractions. Compounds detectable at 350 nm—presumably flavonoids—were observed only in fraction eluted with 40% of methanol (Figure S2), but in other fractions these compounds were absent. The peaks detected in this fraction were not identical with any of the screened flavonoids (including the previously reported apigenin, rutin, and quercetin) [18–20]. The supposed total flavonoid content (peaks at 350 nm), expressed in apigenin (Figure S3), was $0.4670 \pm 0.0146$ mg in 100 g lyophilized powder. The flavonoid content of the analyzed sample was rather low, hence, the supposed antioxidant effect of wheatgrass may be attributed to other compounds than the flavonoids.

A further major goal was to study the effect of wheatgrass juice administration on blood coagulation parameters and its potential interaction with warfarin in rats. The prothrombin time was used for the determination of clotting time. The prothrombin times before treatment and at the end of the treatment are shown in Table 1 (numerical presentation in Table S2). Significant prolongation of the clotting time could be observed in groups treated with warfarin. Interestingly, the treatment with wheatgrass caused 1.6 s prolongation according to the measurement carried out by the mechanical coagulometer, however no significant difference could be established using the optical coagulometer.

**Table 1.** Prothrombin times before and after treatment. The data are presented as mean $\pm$ SD. (WG: wheatgrass; WF: warfarin; WW: wheatgrass + warfarin).

| Group | Prothrombin Time (in Seconds) Determined by Means of Mechanical Coagulometer | | Prothrombin Time (in Seconds) Determined by Means of Optical Coagulometer | |
|---|---|---|---|---|
| | **Before** | **After** | **Before** | **After** |
| WG | $16.48 \pm 0.89$ | $18.07 \pm 1.48$ ** | $19.06 \pm 1.90$ | $19.18 \pm 1.00$ |
| WF | $17.48 \pm 0.62$ | $38.10 \pm 2.27$ ** | $18.23 \pm 0.55$ | $38.06 \pm 3.34$ ** |
| WW | $17.60 \pm 0.96$ | $41.87 \pm 9.47$ * | $18.04 \pm 0.74$ | $44.98 \pm 5.42$ ** |

* $p < 0.05$; ** $p < 0.01$, before vs. after.

Significantly prolonged clotting times were observed in groups receiving warfarin (Figures 1 and 2) compared with those receiving wheatgrass ($p < 0.01$). However, co-administration of wheatgrass seemed to have no effect on the activity of warfarin, since there was no significant difference between groups treated with warfarin (WF) or with combination of warfarin and wheatgrass (WW).

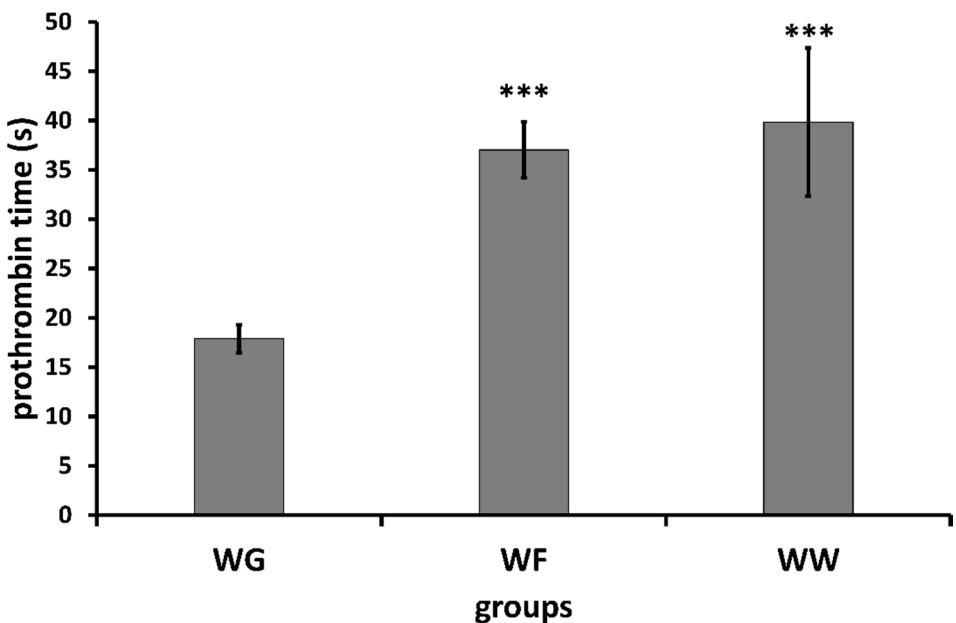

**Figure 1.** Mean $\pm$ SD of prothrombin time determined by means of mechanical coagulometer (WG: wheatgrass; WF: warfarin; WW: wheatgrass + warfarin). *** $p < 0.01$.

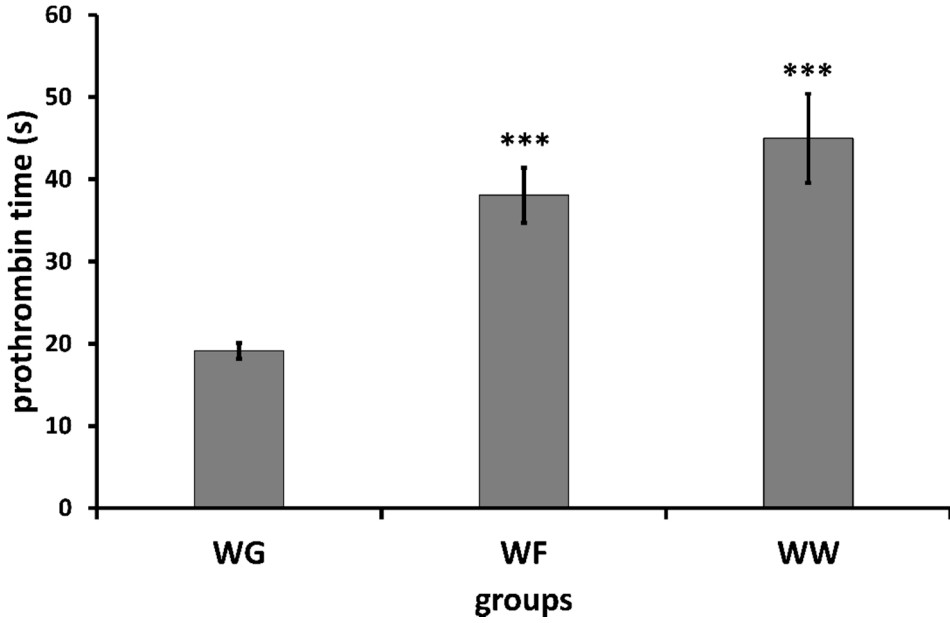

**Figure 2.** Mean $\pm$ SD of prothrombin time determined by means of optical coagulometer (WG: wheatgrass; WF: warfarin; WW: wheatgrass + warfarin). *** $p < 0.01$.

Remarkable clinical symptoms were not recorded in rats during the observation. No spontaneous bleeding was observed, indicating that warfarin was not overdosed (Tables S3–S7).

## 4. Discussion

Although the in vitro cytotoxic effect of wheatgrass was observed on different cancer cell lines, wheatgrass is usually considered to be safe for human use. The main component of extracts is chlorophyll, and no toxic compounds have been reported from this plant. In an animal experiment, wheatgrass extract was practically nontoxic at a dosage of 2000 mg/kg [4]. In clinical studies, the use of wheatgrass has not been associated with significant side effects, only nausea and rashes have been reported [5,37]. In our animal experiment, no clinical symptoms were observed in the animals treated with wheatgrass, this supports the safety of this material.

However, the potential of interactions with oral anticoagulants could not be ruled out based on previous clinical reports. It was reported that wheatgrass was used to treat bleedings and it was claimed that it increased the prothrombin index [6,7].

In this preliminary experiment done in rats, we were curious about the effect of wheatgrass on blood coagulation parameters, the potential interactions between wheatgrass and the most frequently used anticoagulant, warfarin.

The reason for using coagulometers operating based on different principles was to minimize the risk of preanalytical errors. The results obtained by the two coagulometers were concordant, except the slight prolongation of coagulation time measured with the mechanical coagulometer in animals treated with wheat grass. The results presented here suggest that wheatgrass has no univocal effect on prothrombin time and no interaction exists with warfarin, since co-administration of wheatgrass with warfarin did not affect the anticoagulant activity of the latter.

Moreover, our experiments confirmed very low amount of flavonoids present in wheatgrass. None of the previously reported flavonoids (rutin, quercetin, and apigenin) were detected [18–20]. Previous studies on this topic applied non-specific methods for the detection of flavonoids, which may have resulted in the over-estimation of flavonoid content. Further research is needed to clarify whether the low flavonoid concentration we measured is an individual case or generally true for all types of wheatgrass. Based on our results, claims and assumptions based on the supposed presence of flavonoids in wheatgrass warrant special reservations.

Although herbal drugs usually have milder side effects than synthetics, medicinal plants may influence the clinical effects of other medicines by pharmacokinetic or pharmacodynamic interactions. This issue has not been extensively studied in case of the majority of medicinal plants, however, there are some pharmacons that are known to have interaction with several plants. Warfarin, when consumed with danshen (*Salvia miltiorrhiza*) enhanced anticoagulation and bleeding [38], when co-administered with garlic (*Allium sativum*) increased clotting time [39], when consumed with *Ginkgo biloba* bleeding and high blood pressure occurred [40]. Leite and colleagues [31] found 58 different plants that may alter the blood hemostasis and anticoagulation with warfarin and they highlighted the potential risk because of its narrow therapeutic index. In another study [41] only four plants (cranberry, soybean, St John's wort, and danshen) were regarded as highly probable to clinically interact with warfarin. Although wheatgrass is a widely used medicinal plant, its potential interaction with warfarin has not been analyzed in animal experiments so far.

In our study no synergistic or antagonistic interaction was observed after the co-administration of warfarin and wheatgrass. Co-consumption of therapeutic herbals with some prescription drugs often lead to herb-drug interactions, but this depends not only on the administered medicinal plant but also on the composition of the preparation used. The circumstances of extraction (e.g., extracting solvent, temperature, method, drug-extract ratio) may largely influence these interactions. In our experiments, the lyophilizate of unpurified wheatgrass juice was used in order to examine the interaction potential of a product containing all the constituents of wheatgrass that could be obtained by mechanical pressing. Further studies are needed to assess the interaction of other wheatgrass products to have a wider outlook on the interaction potential of this herbal product.

**Supplementary Materials:** The following are available online at https://www.mdpi.com/article/10.3390/scipharm89030039/s1, experimental data and observations are available in Tables S1–S7 and Figures S1–S3. Table S1: Freeze-drying process. Table S2: Database of prothrombin time before and after treatment used for statistical analysis. Table S3–S7: Weight of animals, amount of administered warfarin and wheatgrass on 1–5 day of treatment. Figure S1: HPLC chromatogram of reference compounds ($\lambda = 350$ nm). Figure S2: HPLC chromatogram of fraction eluted with 40% of methanol and apigenin ($\lambda = 350$ nm). Figure S3: Calibration curve (apigenin).

**Author Contributions:** Conceptualization, D.C. and A.S.; formal analysis, M.U., T.K., D.C. and A.S.; funding acquisition, D.C.; investigation, I.T., M.U., T.K., E.H., E.L.-Z., Z.P.Z., A.L., G.O. and D.C.; methodology, I.T., M.U., T.K., E.H., E.L.-Z., Z.P.Z., A.L., G.O., D.C. and A.S.; project administration, D.C. and A.S.; supervision, D.C.; visualization, T.K., D.C. and A.S.; writing—review and editing, D.C. and A.S. All authors have read and agreed to the published version of the manuscript.

**Funding:** This research was funded by the National Research, Development and Innovation Office, grant number OTKA K115796; Economic Development and Innovation Operative Programme, grant number GINOP-2.3.2-15-2016-00012; and János Bolyai Research Scholarship of the Hungarian Academy of Sciences.

**Institutional Review Board Statement:** The experiment conforms to the Guide for the Care and Use of Laboratory Animals published by the U.S. National Institutes of Health (National Institutes of Health publication 85–23, revised 1996), and the regulations of the Hungarian Act No. XXVIII of the year 1998 on protection and care of animals were strictly followed. The study was approved by the Committee on the Ethics of Animal Experiments of the University of Szeged and the Directorate of Food Safety and Animal Health Care, Government Agency of Csongrád County (XXI./151/2013.).

**Informed Consent Statement:** Not applicable.

**Data Availability Statement:** The data presented in this study are available in article and supplementary.

**Acknowledgments:** The authors acknowledge Ákos Bajtel for reviewing the manuscript.

**Conflicts of Interest:** The authors declare no conflict of interest.

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
