# Peer review of "The Effect of Wheatgrass Lyophilizate on Blood Clotting Time in Rats"

_scipharm, doi:10.3390/scipharm89030039_

Round 1
Reviewer 1 Report
The authors have revised the article and provided a point - by - point response my previous comments, improving significantly the quality of the manuscript.
Author Response
Thank you for revising our manuscript.
Sincerely, the authors
Reviewer 2 Report
This is another message of the article. Unfortunately, it should be clearly stated that the suggestions of the reviewers were not taken into account.
Fragments that have been crossed out and re-placed in the same place cannot be considered as an amendment to the text. Applies to:
Line 245-253, chapter 4. Discussion
... "However, the potential of interactions with oral anticoagulants could not be ruled out based on previous clinical reports. It was reported that wheatgrass was used to treat bleedings and it was claimed that it increased the prothrombin index [6.7]. One of the main goals of our experiments was to investigate the effect of wheat grass juice on blood coagulation parameters and its potential interaction with warfarin. The results of the animal experiment presented here suggest that wheatgrass has no univocal effect on prothrombin time and no interaction exists with warfarin, since co-administration of wheatgrass with warfarin did not affect the anticoagulant activity of the latter. Moreover, our experiments confirmed low amount of flavonoids present in wheatgrass .... "
Replaced with Row 253-261 ... "However, the potential of interactions with oral anticoagulants could not be ruled out
based on previous clinical reports. It was reported that wheatgrass was used to treat bleedings and it
was claimed that it increased the prothrombin index [6.7]. One of the main goals of our experiments was to investigate the effect of wheat grass juice on blood coagulation parameters and its potential
interaction with warfarin. The results of the animal experiment presented here suggest that wheatgrass has no univocal effect on prothrombin time and no interaction exists with warfarin, since co-administration of wheatgrass with warfarin did not affect the anticoagulant activity of the latter. Moreover, our experiments confirmed very low amount of flavonoids present in wheatgrass. None of the previously reported flavonoids (rutin, quercetin and apigenin) were detected [18-20]. "...
Such a procedure proves that the authors are not very serious about the reviewer's opinion.
It is also impossible not to respond to the desperate requests of the authors to the editor of the edition for the re-adoption of the article in the still, I repeat, unchanged form, because they are not reluctant to answer the questions regarding the ambiguity and doubts regarding the research described in the manuscript.
The regulations clearly state that the manuscripts are peer-reviewed by at least two reviewers. The authors are required to respond to the indicated comments and include the corrected fragments in the text of the manuscript. I find it highly unethical for the authors to do so. Especially that it refers only to the reviews that do not indicate errors and ambiguities regarding the research carried out. It is the duty of the reviewer to review the manuscript and identify specific errors. As well as pointing out obscure points that only the authors can clarify because they did the research.
Therefore, I stand by my opinion and believe that the article cannot be published in its current form.
Author Response
Dear Reviewer,
we have revised our manuscript, including the discussion part. We hope that this updated version will meet you expectations.
Sincerely, the authors
Reviewer 3 Report
The article ‘’The Effect of Wheatgrass Lyophilizate on Blood Clotting Time in Rats” looks interesting but there are some weak points that should be corrected prior to publication.
First of all, it is not explained nowhere why warfarin was chosen for this experiment since it is not the only drug used for anticoagulant therapy in modern medicine. In the article it is mentioned the “interaction with oral anticoagulants”.
The dosage of wheatgrass used is not clear. Why 7fold increase of human dose was used? Is the calculations correct? If we multiply 25.5*7 we will obtain 178.5. Such correlation of doses seems to be incorrect since rodents have a much quicker metabolism.
The dosing process should be written clearer because it is not obvious that a single animal obtained proper drug dose.
Line 150. “All animals received 4 g cookie dough/kg b.w. orally” how many WF or WG were inside? It should clarify. How it was divided for rats since their body weight is about 250 g? Why this administration way was used? Why oral gavage was not applied as more accurate one?
Line 154. Body weight.
Line 156. “and given to each animal individually, also in the morning hours between 10 and 11 AM while they were left in their home” what it is about?
Table 1. No units of prothrombin time are given. Also, there is no statistical analysis of obtained data.
Usually in such experiments control group should be used in the present article there is no such one.
Fig. 1 - 4. Given * p< 0.05 is not stated vs which group.
Moreover, all these figures 1-4 seems to be excess since all necessary information is present in the table 1. Why authors give so much doubled information?
It is not clear why authors compare two coagulometers. Is it really important? Seems not.
Discussion part looks very poor.
Author Response
Dear Reviewer,
thank you for the careful revision of our paper. Please find our answers below. Sincerely, the authors
First of all, it is not explained nowhere why warfarin was chosen for this experiment since it is not the only drug used for anticoagulant therapy in modern medicine. In the article it is mentioned the “interaction with oral anticoagulants”.
In our experiment warfarin was chosen because it is the most extensively used oral anticoagulant for the prevention and treatment of thromboembolic complications in cardiovascular diseases (Choi et al., 2011). See: Line 95-97
The dosage of wheatgrass used is not clear. Why 7fold increase of human dose was used? Is the calculations correct? If we multiply 25.5*7 we will obtain 178.5. Such correlation of doses seems to be incorrect since rodents have a much quicker metabolism.
We thank for this very useful comment, there was a mistake in the description of the calculation. The average daily dose was multiplied according to the FDA guideline; however the description was incorrect.
The dosing process should be written clearer because it is not obvious that a single animal obtained proper drug dose.
Line 150. “All animals received 4 g cookie dough/kg b.w. orally” how many WF or WG were inside? It should clarify. How it was divided for rats since their body weight is about 250 g?
The basic recipe for sugar cookie dough included: 55% plain flour, 20% castor sugar and 25% water. Animals received plain cookie dough with wheatgrass (WG), or warfarin (WF) or the combination of wheatgrass and warfarin (WW) according to the measured body weight. For example, if in the morning the body weight of a rat in WW group was 252 g, it received (0.4 mg/kg b.w. =) 0.1008 mg warfarin and (127.5 mg/kg b.w. =) 32.13 mg wheatgrass in 4 g cookie dough/kg b.w.. See: Line 168-169.
Line 154. Body weight.
Thank you, corrected.
Why this administration way was used? Why oral gavage was not applied as more accurate one?
Instead of traditional gavage technique oral administration of cookie dough was applied in order to cause the least stress and harm to the animals and to model human exposure the most objectively. In our previous experiment (Kiss et al., 2017) this administration way of unpleasant tasty drug was successful. Rats ate the sweet cookie balls with pleasure, and voluntarily. The researcher always made sure that the whole ball was eaten. Furthermore, oral gavage treatment is potentially dangerous to the animal, a common complication with gavage involves perforation of the esophagus or stomach. See: Line 160-161.
Line 156. “and given to each animal individually, also in the morning hours between 10 and 11 AM while they were left in their home” what it is about?
During the study, the body weights of animals were measured every morning. According to the measured body weight individual sugar cookie balls were prepared and then immediately given to the rats as they were rather hungry after overnight fasting. We wanted to ensure a predictable home-cage environment even during eating to avoid they become stressed and refrain from eating the balls. See: Line 164.
Table 1. No units of prothrombin time are given. Also, there is no statistical analysis of obtained data.
Missing data were added.
Usually in such experiments control group should be used in the present article there is no such one.
In this experiment we aimed to assess the effect of wheatgrass on blood clotting and the potential wheatgrass-warfarin interactions in rats. Control group is not necessary to evaluate this effect. Furthermore, we want to reduce the number of animals used in the experiment according to the Ethical Guidelines for the Use of Animals in Research.
Fig. 1 - 4. Given * p< 0.05 is not stated vs which group. Moreover, all these figures 1-4 seems to be excess since all necessary information is present in the table 1. Why authors give so much doubled information?
Figures were eliminated and marking of significance was improved.
It is not clear why authors compare two coagulometers. Is it really important? Seems not.
We had the opportunity to carry out these experiments using two instruments with different operating principles. Although it is no necessary to measure coagulation parameters by two methods, in our opinion these consistent results reassure the validity of our findings.
Discussion part looks very poor.
Discussion was amended and improved.
Round 2
Reviewer 2 Report
The manuscript is slightly improved. The main changes concern the Discussion chapter. Unfortunately, in the remaining chapters, large sections of the text were first deleted and then re-pasted. If that's the manuscript improvement, the authors don't seem to know how to do it. This is amazing because the authors have done the research and they should know the answers to the questions. At the same time, they should be able to explain phenomena that raise doubts. The role of the reviewer is to care for the best name of the journal, therefore, unfortunately, I still believe that the manuscript in its current form cannot be published in Sci. Pharm.
Author Response
Dear Reviewer,
first of all, we are sorry for the misunderstanding. We were not sure that Reviewer 2 of scipharm-1005775 is identical with Reviewer 2 of scipharm-1054276, so in the first round of revision we did not answer the questions raised during the previous submission.
Here we provide detailed answers for 3 revision rounds.
Sincerely, the authors
Reviewer #2's comments for the scipharm-1005775 in November 2020
The authors only referred to some of the suggestions. They corrected the title of the article. Information on the freeze-drying process was given by including data in a table that is not included in the text.
Information on the freeze-drying process can be found in the supplementary material (Table S1)
The description of the phase gradient in the determination of flavonoids remained almost unchanged. Underline "Gradient elution was applied (eluent B 20–40–55–70–70–20–20% in 1–20–24–27–28–28.5–34 minutes)". Only the time has been increased from 33 minutes to 34 minutes. In my opinion, it does not change anything and remains little understood.
The description presented here allows the reproduction of the chromatographic process by experts in the field. All the necessary details can be found in the manuscript (column, eluent composition gradient, flow, wavelength, injection volume, temperature). What is presented in the section 2.3, is totally in accordance with the structure and information content of similar papers.. We do not understand what is missing according to the Reviewer’s opinion.
Unfortunately, there is still no supplement and no reliable confirmation of the sense of the research carried out. Lines 177-184: describe problems with the determination of flavonoids in freeze-dried juice. However, it is not clear whether the juice was additionally enriched with these compounds or whether it was enough to use a different solvent for extraction.
This is clearly described in section 2.2.: “During the preliminary HPLC experiments lyophilized wheatgrass juice was dissolved in methanol. For detailed HPLC analysis, 10 g of lyophilized wheatgrass juice powder (equivalent to 238 g of fresh wheatgrass juice) was fractioned on a 70 g polyamide column”. Everything is disclosed in this section, what is necessary to understand and reproduce these experiments.
It proves, first of all, low knowledge of the product used for research. These errors also result from the lack of awareness of the impact of the lyophilization process and subsequent powder preparation parameters, which may modify the composition of the plant matrix. The authors gave only an enigmatic answer: "The enrichment of the flavonoids was done by fractionation on polyamide. This method allows the enrichment of the flavonoids with fractions with high MeOH content. And it just happened: although no flavonoid was detected in the crude solution, the flavonoid was not detected in 40% of the fraction. MeOH. "
We reject this subjective and unworthy claim. We would be grateful if the Reviewer would abstain from such remarks.
The rationale of enrichment was to increase the concentration of the flavonoids to achieve a level above the limit of detection and quantification Using polyamide for this purpose is a common practice in phytochemical analysis. In this experiment, we aimed to detect some common flavonoids (among these some compounds that were previously reported from the plant). Although none of these was detectable after enrichment, in one fraction some compounds were detectable with the typical UV absorption of flavonoids.
The question remains whether the content of apigenin. in the extract can be treated as the sum of flavonoid compounds? You can be sure not.
The AUC-s of compounds detected with typical flavonoid-like UV-absorption were summarized and expressed in apigenin content. This is a common and accepted practice.
Lines 123-126: “Using a 6 point calibration curve, the contents of apigenin, apigenin-7-O-glycoside, eupatorin, hesperidin, hyperoside, kaempferol, luteolin, rutin and quercetin were measured. The total content of flavonoids was expressed. in apigenin. ”From this description it is difficult to deduce what the authors meant! How was the content of these flavonoids measured? From the information presented, it should be concluded that the patterns are apigenin, apigenin-7-O-glycoside, eupatorin, hesperidin, hyperoside, kaempferol, luteolin, rutin and quercetin. A standard curve was prepared and plotted for each standard at different concentrations (6), on the basis of which the content of identified flavonoids in the sample was calculated. Assuming that their sum determines the total content in the test product / sample.
We developed an analytical method to measure some common flavonoids (among these some compounds that were previously reported from the plant). Then we used this method to identify and measure these compounds in wheatgrass. This is a reasonable and pragmatic approach for the flavonoid content. But since none of the marker compounds was detectable, we switched to a method that allows total flavonoid content determination (as presented above).
Throughout the text, the authors use very large mental shortcuts. After all, the article is going to reach different readers. The authors explain that "Before analyzing the wheatgrass sample, we developed a method for the qualitative and quantitative determination of flavonoids. For all standards, we established calibration curves based on 6 points. However, there were no detectable flavonoids in the raw sample (methanol solution), only in the enriched fraction (eluted 40% MeOH) We have revised the manuscript to make this aspect more understandable. " As it results from the presented data, the developed method is inadequate to the indicated matrix. I still emphasize that there are no preliminary data confirming the presence of these compounds.
These issues are answered above.
The Results chapter: does not include the most important achievements and experiments performed. Authors casually. They described selected passages, making the data difficult to interpret.
This section was improved, we hope, it meets the expectation of the Reviewer.
The suggested expansion of the discussion was not carried out. The discussion chapter, based on 5 references, is not the strong point of the article. The authors do not discuss the obtained results with the literature data.
This section was improved too, we hope, it meets the expectation of the Reviewer.
To sum up, the correction of the article and the enigmatic explanations of the authors do not allow the publication of the presented evidence in Scientia Pharmaceutica. I suggest rejecting the article as it has serious flaws, additional experiments are needed and the research is poorly done.
Reviewer 2’s comments for the scipharm-1054276:
Round 1:
This is another message of the article. Unfortunately, it should be clearly stated that the suggestions of the reviewers were not taken into account.
Unfortunately we are not aware what kind of suggestions should be taken into account in this 1st round of revision.
Fragments that have been crossed out and re-placed in the same place cannot be considered as an amendment to the text. Applies to:
Line 245-253, chapter 4. Discussion
... "However, the potential of interactions with oral anticoagulants could not be ruled out based on previous clinical reports. It was reported that wheatgrass was used to treat bleedings and it was claimed that it increased the prothrombin index [6.7]. One of the main goals of our experiments was to investigate the effect of wheat grass juice on blood coagulation parameters and its potential interaction with warfarin. The results of the animal experiment presented here suggest that wheatgrass has no univocal effect on prothrombin time and no interaction exists with warfarin, since co-administration of wheatgrass with warfarin did not affect the anticoagulant activity of the latter. Moreover, our experiments confirmed low amount of flavonoids present in wheatgrass .... "
Replaced with Row 253-261 ... "However, the potential of interactions with oral anticoagulants could not be ruled out based on previous clinical reports. It was reported that wheatgrass was used to treat bleedings and it was claimed that it increased the prothrombin index [6.7]. One of the main goals of our experiments was to investigate the effect of wheat grass juice on blood coagulation parameters and its potential interaction with warfarin. The results of the animal experiment presented here suggest that wheatgrass has no univocal effect on prothrombin time and no interaction exists with warfarin, since co-administration of wheatgrass with warfarin did not affect the anticoagulant activity of the latter. Moreover, our experiments confirmed very low amount of flavonoids present in wheatgrass. None of the previously reported flavonoids (rutin, quercetin and apigenin) were detected [18-20]. "...
Such a procedure proves that the authors are not very serious about the reviewer's opinion.
I suppose this remark refers to the use of track changes function, which allows to follow all the changes in the manuscript (also when a piece of the text is replaced). We try to answer all the questions, but in this case this was only a simple reorganization of the text. If there is any specific question, we will try to improve the manuscript accordingly.
It is also impossible not to respond to the desperate requests of the authors to the editor of the edition for the re-adoption of the article in the still, I repeat, unchanged form, because they are not reluctant to answer the questions regarding the ambiguity and doubts regarding the research described in the manuscript.
We do not agree. Several amendments were carried out during the revision process. If there are any specific comments, we will try to answer them.
The regulations clearly state that the manuscripts are peer-reviewed by at least two reviewers. The authors are required to respond to the indicated comments and include the corrected fragments in the text of the manuscript. I find it highly unethical for the authors to do so. Especially that it refers only to the reviews that do not indicate errors and ambiguities regarding the research carried out. It is the duty of the reviewer to review the manuscript and identify specific errors. As well as pointing out obscure points that only the authors can clarify because they did the research.
Therefore, I stand by my opinion and believe that the article cannot be published in its current form.
Thank you for your opinion. We will try to asnwer the specific remarks.
Reviewer 2’s comments:
Round 2:
The manuscript is slightly improved. The main changes concern the Discussion chapter. Unfortunately, in the remaining chapters, large sections of the text were first deleted and then re-pasted. If that's the manuscript improvement, the authors don't seem to know how to do it. This is amazing because the authors have done the research and they should know the answers to the questions. At the same time, they should be able to explain phenomena that raise doubts. The role of the reviewer is to care for the best name of the journal, therefore, unfortunately, I still believe that the manuscript in its current form cannot be published in Sci. Pharm.
We will try to answer if there are specific comments. We did not find any remarks concerning “phenomena that raise doubts”, therefore we find unreasonable this negative opinion.
Reviewer 3 Report
There are still some points that should be improved:
Line 160. I disagree with the statement about harm caused by oral gavage it is a safe method if done by trained researcher.
Please indicate the time during which an animal ate a ball with tested compounds.
Table 1 - "*" is usually putted after SD. Also it is very strange the significant difference between WG before and after in case of mechanical coagulometer and it is not discussed anywhere. The usage of two coagulometers also didn't described/discussed in the text.
Line 250 - "Significantly prolonged clotting..." no such analysis is presented in the proper table.
Author Response
Dear Reviewer,
thank you for carefully revisiong our manuscript. Your remarks helped us to improve the paper. Please find our answers below.
Line 160. I disagree with the statement about harm caused by oral gavage it is a safe method if done by trained researcher.
Please indicate the time during which an animal ate a ball with tested compounds.
In our previous experiments a number of different chemical agents were
administered orally by gavage. During the last years in our subchronic
experiments we tried a new method of oral treatment which was also used
by other researchers (Walker et al., 2012) and it is in accordance with
the ethical guidelines for animals used in research (to cause less
stress/pain/harm to the animals).
Rats ate the cookie dough with pleasure, the whole cookie dough was
eaten within 1 minute. We concluded that this cookie dough dosing method
is a significantly less stressful alternative to oral gavage.
Reference: Mary K. Walker, Jason R. Boberg, Mary T. Walsh, Valerie Wolf, Alisha
Trujillo, Melissa Skelton Duke, Rupert Palme, Linda A. Felton,
A less stressful alternative to oral gavage for pharmacological and
toxicological studies in mice, Toxicology and Applied Pharmacology,
Volume 260, Issue 1, 2012, Pages 65-69.
Table 1 - "*" is usually putted after SD.
Thank you, corrected.
Also it is very strange the significant difference between WG before and after in case of mechanical coagulometer and it is not discussed anywhere. The usage of two coagulometers also didn't described/discussed in the text.
This is discussed in the manuscript now in short.
Line 250 - "Significantly prolonged clotting..." no such analysis is presented in the proper table.
This was presented in 2 figures that were deleted in a previous step of te revision according to the proposal of one of the reviewers. Now figures 1 and 2 are again part of the manuscript, conformimng this statement.
Round 3
Reviewer 2 Report
The authors addressed some of the suggested suggestions and modified the article to some extent. The revised version contains minimal corrections which, in my opinion, do not allow the article to be accepted for publication. The indicated issues that raise doubts should be clarified by the authors.
Author Response
since in round 2 and 3 no substantial comments were mady by Reviewer 2, we were not able to follow them